# Concomitant Hepatectomy and Atrial Thrombectomy under Cardiopulmonary Bypass versus Staged Hepatectomy in the Treatment for Hepatocellular Carcinoma with Large Right Atrial Tumor Thrombi

**DOI:** 10.3390/jcm11082140

**Published:** 2022-04-12

**Authors:** Wen-Shan Chao, Ching-Hui Shen, Shao-Ciao Luo, Feng-Hsu Wu, Hao-Ji Wei, Chu-Leng Yu, Cheng-Chung Wu, Yun Yen, Fang-Ku P’eng

**Affiliations:** 1Department of Surgery, Ministry of Health and Welfare Feng Yuan Hospital, Taichung City 42055, Taiwan; nicolechao0706@gmail.com; 2Departments of Anesthesiology, Taichung Veterans General Hospital, Taichung City 40705, Taiwan; forscipaper@gmail.com; 3Department of Anesthesiology, Faculty of Medicine, National Yang-Ming University, Taipei City 112304, Taiwan; 4Departments of Surgery, Taichung Veterans General Hospital, Taichung City 40705, Taiwan; ytppytpp@hotmail.com (S.-C.L.); hegsvghtc@gmail.com (C.-C.W.); fkpeng@me.com (F.-K.P.); 5Department of Nursing, Hung Kuang University, Taichung City 433304, Taiwan; 6Departments of Cardiovascular Surgery, Taichung Veterans General Hospital, Taichung City 40705, Taiwan; weihaoji@vghtc.gov.tw (H.-J.W.); julingyu@gmail.com (C.-L.Y.); 7Department of Surgery, Faculty of Medicine, National Yang-Ming University, Taipei City 112304, Taiwan; 8Cancer Translation Research Center, Taipei Medical University, Taipei City 106, Taiwan; yyen@tmu.edu.tw

**Keywords:** hepatocellular carcinoma, large right atrial tumor thrombi, cardiopulmonary bypass, atrial thrombectomy, hepatectomy

## Abstract

(1) Background: Hepatocellular carcinoma (HCC) with a large right atrium tumor thrombus (RATT) is a rare and critical presentation. Emergency hepatectomy and thrombectomy under cardiopulmonary bypass (CPB) is life-saving and potentially curative. The aim of this study is to propose an appropriate approach for this condition. (2) Methods: In period A (1998 to 2010, *n* = 7), hepatectomy and thrombectomy were concomitantly performed, and staged hepatectomy was performed in period B (2011 to 2018, *n* = 17). (3) Results: The median overall survival time (MOST) in the published studies was 14 months. Moreover, the blood loss, blood transfusion rate, length of ICU stays, and hospital costs were significantly reduced in period B. The MOSTs of patients in period A (*n* = 6) and period B (*n* = 17) were 14 vs. 18 months (*p* = 0.099). The median disease-free survival times (MDFTs) in period A (*n* = 6) and period B (*n* = 15) were 8 vs. 14 months (*p* = 0.073), while the MOSTs in period A and period B were 14 vs. 24 months (*p* = 0.040). (4) Conclusions: Emergency thrombectomy under CPB and staged hepatectomy 4–6 weeks later may be an appropriate approach for HCC with large RATT. However, the optimal waiting interval requires further investigation.

## 1. Introduction

Hepatectomy remains one the most common curative modality to treat hepatocellular carcinoma (HCC). HCC usually spreads intra-hepatically via the portal veins [1]. Cases where HCC invades via the hepatic vein (HV) to the inferior vena cava (IVC) and right atrium (RA) are rare. The incidence was found to be 2–3.8 percent, and the consequences are grave [1,2,3,4,5,6,7,8,9,10,11,12,13,14,15,16,17,18,19,20,21,22,23,24]. If untreated, patients will die of extensive intrahepatic and systemic metastases within two to three months [2,3]. When the RA tumor thrombi (RATT) are large enough to occupy a large part of the lumen (Figure 1A), it will cause acute circulation obstruction leading to sudden death (“hole-in-one thrombus” effect [4]). 

Sakamoto and Nagano classified this condition as type III, intracardiac (TT entering the RA) [5]. Emergency hepatectomy with thrombectomy under cardiopulmonary bypass (CPB) is a life-saving and the only potentially curative modality [1,12,13,14,15,16,17,18,19,20,21,22,23,24,25]. However, there are side effects from systemic heparinization in CPB as hepatectomy with thrombectomy is complex and challenging procedure. Since first reported by Onitsuka et al. in 1990 [26] regarding a successful concomitant procedure of left hepatectomy and RA thrombectomy under CPB in treating such patients, [16] less than 50 articles have been published. 

Due to the side effects of CPB, the combination with hepatectomy is formidable. In 2011, a 41-year-old woman was referred to us due to a large lipiodolized HCC with RATT that presented with cardiogenic shock. The patient underwent emergency thrombectomy under CPB and subsequential extracorporeal membrane oxygenation (ECMO) therapy and due to moribund status, and then a right hepatectomy was performed 4 weeks after recovery from CPB. We have adopted this strategy since then.

Iemura [18] and Saïsse J. [20] had proposed a staged hepatectomy concept for this condition; however, no clinical investigations have been obtained in practice. A staged hepatectomy with a waiting interval was originally applied to selected patients, and it produced a better outcome after hepatectomy for simultaneous colorectal cancer liver metastasis (CCLM) [27,28], liver hilum cholangiocarcinoma [27,28,29], down-staged bi-lobar multiple CCLM [27,28,29] and large or ruptured HCC after trans-arterial chemoembolization (TACE) [30,31].

Herein, we conducted a retrospective review of the last 21 years on 24 HCC patients admitted for curative intends hepatectomy and thrombectomy under CPB. The aim of this study is to appraise the strategy changes for potentially curative HCC with RATT.

## 2. Materials and Methods

### 2.1. Preoperative Assessments for Hepatectomy of HCC with RATT

The resect-ability of HCC with RATT patients was determined by their general condition, tumor extension by image studies and liver function reserve indocyanine-green 15-min retention rate (ICGR15) value (the clinicopathological characteristics are listed in Table 1) according to a modified Makuuchi’s criteria [31,32,33,34], where all information must be obtained within 2 to 3 h at admission for the following emergency operation. From 1998 to 2018, 19 of 1962 patients with primary HCC, and 5 of 528 patients with recurrent HCC associated with large RATT (≥3 cm) met the inclusion criteria. They all presented with ascites, exertional dyspnea, orthopnea, leg edema and engorged jugular veins [35]. The blood transfusion policy followed our previous article [36].

### 2.2. Intra- and Post-Operative Assessments of Patients (n = 7) in Period A (1998–2010)

During period A, hepatectomy and RA thrombectomy under CPB were concomitantly performed as an emergency schedule. Five of seven patients underwent hepatectomy followed by thrombectomy, and the other two had thrombectomy first. The following surgical details were conducted in our previous published articles. In addition, the cell-saver apparatus was not used. The abdomen was opened using a “right upper J-shaped” or “Mercedes–Benz” incision [31,32,33,34]. After Kocher’s maneuver, the infra-hepatic suprarenal IVC and hepatoduodenal ligament were exposed and taped.

After cholecystectomy with assisted intraoperative ultrasound, the liver parenchyma was transected using the Kelly clamp-crush method under intermittent Pringle maneuvers (15 min clamping and 5 min de-clamping) until reaching supra-hepatic IVC [31,32,33,34], which was fully exposed by longitudinal splitting of the diaphragm.

The priming fluid in CPB consisted of lactated Ringer’s solution 1500 mL, mannitol 150 mL and 20% albumin 100 mL. After median sternotomy and pericardiotomy, the superior vena cava (SVC) and ascending aorta (AA) were individually taped, and cardiac cannulations were set with venous line through the femoral vein and SVC.

Intravenous bolus injection of heparin (300 IU/kg body weight) was used when the activated clotting time (ACT) was over 400 s (normal, <120 s). CPB was initiated after clamping the hepatoduodenal ligament, SVC, infra-hepatic suprarenal IVC and AA. Thereafter, cardioplegia was injected into the aortic root to arrest heartbeats. The RA and cranial portion of IVC were opened (Figure 1B). After extraction of the thrombi (Figure 1C), the resected liver and thrombi were removed en bloc.

The wounds over diaphragm, RA and IVC were finally closed. After defibrillation, the heartbeat resumed its beating and the CPB terminated. Protamine (a 1–1.5 mg/100 IU total dose of heparin) was dripped intravenously to reverse the anticoagulant effect of heparin followed by placing drainage tubes and closing wounds of laparotomy and sternotomy. After surgery, the patient was sent to the intensive care unit (ICU), with correction of Heparin-related coagulopathy and a weaning mechanical ventilator.

### 2.3. Intra- and Post-Operative Assessments of Patients (n = 17) in Period B (2011 to 2018) 

During period B, hepatectomy and thrombectomy under CPB were conducted as two separate operations.

### 2.4. First Operation

This operation (Figure 1D) was similar to the thrombectomy procedures in period A but without cholecystectomy, liver mobilization, hepatectomy and opening of the diaphragm. Thrombectomy was performed under a continuous Pringle maneuver and CPB. The thrombi at RA, IVC and the orifices of major HV (right, middle and left HV and large right inferior HV) were completely removed. The postoperative assessments were similar to period A.

### 2.5. Second Operation (Staged Hepatectomy)

Patients were re-admitted approximately 4–6 weeks after thrombectomy and similar preoperative assessments. Congested intraabdominal organs and dilated retroperitoneal collateral vessels disappeared during the second operation (Figure 1E). Hepatectomy was performed under intermittent Pringle’s clamping using the low CVP policy. The details of the post-hepatectomy assessments were similar to those described in our earlier reports [31,32,33,34,35].

Due to the failed Pringle maneuver as a result of severe adhesions of the hepatoduodenal ligament and retroperitoneum, thrombectomy was achieved under deep hypothermic (16 C) total circulation arrest on one patient in each period (31 min in period A and 38 min in period B).

Postoperative morbidity and mortality were defined as complications and death that occurred within 90 days or during the same hospitalization. The complication severity was graded using the Clavien–Dindo classifications [37].

### 2.6. Long-Term Follow-Up

Patients were followed up on a monthly basis at the outpatient clinic for the first 12 months after discharge from the hospital and every two to three months thereafter.

### 2.7. Statistical Analyses

The clinicopathological characteristics (Table 1) as well as the early and long-term postoperative results of the two periods (Table 2) were compared. Continuous variables are presented as the median (range) and were compared using the Mann–Whitney U test. For period B, some variables are shown as the sum of both operations. The frequencies were compared using Fisher’s exact test or Pearson’s *χ*^2^ test as appropriate.

The durations of postoperative disease-free survival (DFS) and overall survival (OS) were calculated starting from the day of first admission until cancer recurrence or patient death. They were estimated with the Kaplan–Meier life-table method and compared using the log-Rank test. The statistical significance was set at *p* < 0.05.

## 3. Results

In period B, the preoperative ICG R15 values before the second operation (13.2%; range 6.1–40.6%) tended to be lower than those before the first operation (18.6%; range 7.9–49.4%) (*p* = 0.075). Two patients did not receive staged hepatectomy because of multiple lung metastases and peritoneal metastases on each patient within the waiting period.

No significant differences in all the clinicopathological variables were found; however, the amount of blood loss, blood transfusion rate, length of ICU stays and hospital costs were greater in period A (Table 2). One patient in period A died of overlooked secondary abdominal compartment syndrome (SACS) within 7 days.

No patient had recurrent IVC or RA tumor thrombi before the second operation in period B. The follow-up durations until 2019 of period A were 16 (6–26) months and 21 (4–55) months in period B. We found that 15 of 21 patients (71.43%) developed multiple lung metastases. Recurrent HCCs were treated by TACE (*n* = 4), oral sorafenib (available in our hospital after 2008, *n* = 7) [6] or supportive care (*n* = 4).

Figure 2 demonstrates that the median overall-survival times (MOSTs) of patients in period A (*n* = 6) and period B (*n* = 17) were 14 vs.18 months (*p* = 0.099). Figure 3 shows that, excluding one operative death in period A and two patients without staged hepatectomy in Period B (due to disease progression), the median disease-free survival times (MDFTs) (Figure 3A) in period A (*n* = 6) and period B (*n* = 15) were 8 vs. 14 months (*p* = 0.073) while the MOSTs (Figure 3B) in period A and period B were 14 vs. 24 months (*p* = 0.040).

## 4. Discussion

HCC with RA tumor thrombi and circulation obstruction is a unique and life-threatening presentation. In autopsy cases, the incidence was 4.1% [23]. We accepted patients referred nationwide, and we still only collected 24 cases over the last 21 years. We have collected and analyzed the greatest patient number in one institution, which is the only public government-supported medical center in central Taiwan. By our observation, we found similar long-term outcomes in period A and B and also in published articles but a better post-hepatectomy OS in period B.

Treatments for these patients include medical treatments [6,7], non-resection therapies [8,9], thrombectomy without hepatectomy [10,11] and hepatectomy and thrombectomy with or without cardiopulmonary bypass (CPB) [1,12,13,14,15,16,17,18,19,20,21,22,23,24,25,38,39,40]. Standard treatment protocols are difficult to establish due to the limited experience in treating this disease.

If the thrombi only protrude into a small part of the RA lumen, hepatectomy with thrombectomy can be achieved using the technique of total hepatic vascular exclusion by clamping of the infra-hepatic IVC, the caudal part of the RA wall and the hepatoduodenal ligament without the need of CPB [13,14].

When tumor thrombi are larger (≥3 cm) and markedly occupied the RA lumen (Figure 1), the only curative and life-saving surgical option [13,14,15,16,17,18,19,20,21,22,23,24,25,26,27,28,29,38,39,40] to relieve the hole-in-one thrombus effect [4] is hepatectomy and thrombectomy under CPB [15,16,17,18,19,20,21,22,23,24], as with the previously mentioned 41-year-old female patient. For such considerations, Chu et al. [10] and Tsai et al. [11] only performed life-saving thrombectomy under CPB to alleviate circulatory obstruction and leave the liver tumor intact. Coagulopathy in both CPB and hepatectomy is a very complicated issue [41].

Aggressive anticoagulation is an indispensable part of CPB. Large doses of heparin may cause blood clot lysis and recurrent bleeding from the operative fields where bleeding sites have once been sealed, which are called heparin rebounding phenomena. These phenomena increase the blood transfusion and operative time and considerably increase the operative morbidity and mortality rates [15,16,17,18,19,20,21,22,23,24,25,39,40,41]. In our study period A, the operative bleeding exceeded 10 L in both approaches. Iemura et al. [18] recommended staged hepatectomy performed separately after 3 weeks.

In the past, most surgeons favored hepatectomy before thrombectomy under total hepatic vascular exclusion and CPB [15,16,17,19,21,22,23,24,25,39,40], which can reduce the CPB time and result in no excessive bleeding as well as specimen removal en bloc. On the other hand, in our experience, this can result in extensive bleeding due to blood from the cut surfaces of the liver and wide intra-abdominal raw surfaces due to the (1) venous obstruction by thrombi and (2) coagulopathy by systemic heparinization in CPB.

Moreover, some surgeons prefer thrombectomy first to avoid cancer cells dislodging and spreading from the IVC or RA thrombi during liver mobilization and transection in hepatectomy procedures [18,20,42]. However, in our experience, the massive blood loss because of the congest retroperitoneal collateral vessels and swollen liver do not disappear quickly, and the heparin rebounding phenomena takes at least 3 h to ease after protamine infusion. Iemura et al. [18] and Saïsse et al. [20] recommended that hepatectomy should be performed after thrombectomy under CPB.

Based on the lessons learned from period A and in 2011, one 41-year-old woman who happened to admit for a moribund state had survived urgent thrombectomy under CPB (Figure 1D) and on ECMO lasting for 4 h. She had staged hepatectomy 4 weeks later and the congested retroperitoneal collateral vessels and swollen liver disappeared during the second operation (Figure 1E). We conducted this method in period B with the complex procedures on two separate operations. The extensive operation required the team work of an experienced anesthesiologist and expert surgeons (specialized in abdominal and cardio-thoracic surgeries). This operation involved a wide field of experts in hepatectomy for HCC.

Some important issues should therefore be addressed as follows:

First, the area of intra-abdominal dissection should be kept to the absolute minimum only attempting to tape the hepatoduodenal ligament and infra-hepatic suprarenal IVC in the first operation to reduce postoperative adhesions and to decrease blood oozing from a wider dissected raw surface during thrombectomy. In 2016, Pesi et al. [39] proposed CPB with routine hypothermic cardiocirculatory arrest to decrease the amount of blood loss in patients of HCC with tumor thrombus extending to hepatic or caval veins. Tsai et al. [11] hypothesized that prolonged deep hypothermic cardiocirculatory arrest may cause irreversible cerebral hypoxia, and thus we suggest thrombectomy under deep hypothermia (16 °C) total circulation arrest only if the Pringle maneuver fails to perform.

Second, because the short distance of major HV orifices and RA (usually <2 cm), the field of total thrombectomy is readily visible from the inner surface of the atriotomy and IVC venotomy wound. The thrombi in RA, IVC and major HV orifices rarely adhere to the intima of these vessels, and hence the thrombi can be completely extracted with ease (Figure 1C). The thrombi in HV should be removed as far from the HV orifice as possible. This may avoid recurrent thrombi during the waiting period prior to the second operation.

Third, complications after the first operation and side effects of CPB should be treated in a timely manner. Among them, SACS—a severe and easily overlooked side effect of CPB—is potentially fatal if not treated properly [41]. This side effect was never reported in the past, as it typically occurs after giving a large amount of fluid supplements in CPB. This syndrome can be managed by Silastic silo without early forceful closing of the abdominal wound [43]. After a single death occurred in period A, another three patients (one in period A and two in period B) were successfully saved in this way.

After completing the entire surgical protocol, complications from the first operation in period B can be overcome (Table 2), including decreased blood loss and transfusion rates. Therefore, staged hepatectomy may prolong survival in HCC with large RATT. Staged hepatectomy may also help in curative resection in RATT; however, further cases are still required.

There are the limitations of our present study:

First, because of the rarity of HCC with RA tumor thrombi, our patient number was small. The results may not truly reflect the picture of this advanced HCC. Moreover, this was a retrospective historical comparison. Although the results were better in the more recent period (B), it is possible that the beneficial effects observed were related, at least in part, to improvements in surgical techniques, operative equipment, anesthesiology, perioperative assessments and the development of oral sorafenib. The true benefits of our proposed treatment should be reevaluated using a well-designed prospective randomized trial and on more patients, perhaps in collaboration with investigators overseas.

Second, the optimal waiting interval between thrombectomy under CPB and staged hepatectomy remains unclear. Our interval of staged hepatectomy (4–6 weeks) was based on results from our previous experience on neoadjuvant TACE for large HCC [31] and on the recommendations of experts on the treatment of ruptured HCC [30]. However, our experiences illuminated the results of hepatectomy performed after recovery from CPB. The optimal waiting interval requires further investigation.

A substantial portion of our patients (46.7%) had lung metastasis during cancer recurrence, which is consistent with other published articles. Occult metastasis had likely occurred at the time of hepatectomy and thrombectomy. It is imperative that further studies on the early detection and therapies of occult lung metastasis in such patients should be performed. Recently, new treatments and immunotherapies have been developed, including hepatic arterial infusion chemotherapy, concurrent radiotherapy, systemic chemotherapy and target therapy, such as sorafenib and Regorafenib. Regarding immunotherapy, since 2017, the U.S. Food and Drug Administration (FDA) approved nivolumab for HCC that was previously treated with sorafenib. Immunotherapy has also been combined with targeted agents, such as tyrosine kinase inhibitors, radiotherapy and TACE; however, the objective response rate is still relatively low [44,45].

Despite these limitations, Matsukuma [46] reported 13 patients with RATT whose median OS was 17.4 months; however, only 3 of 13 underwent CPB. In contrast, all the patients in our study underwent CPB, and for staged hepatectomy, our results indicated that staged hepatectomy after thrombectomy under CPB could result in a simplified post-operative course, shortened ICU stay and operation time and prolonged survival time for patients. We also are the first article to analyze the benefits in hospital costs, and the costs for separate admissions were lower than for one admission.

## 5. Conclusions

Emergency thrombectomy under CPB and staged hepatectomy 4–6 weeks later may be an appropriate approach for HCC with large RATT. However, the optimal waiting interval and therapies for occult lung metastasis need to be further investigated.

## Figures and Tables

**Figure 1 jcm-11-02140-f001:**
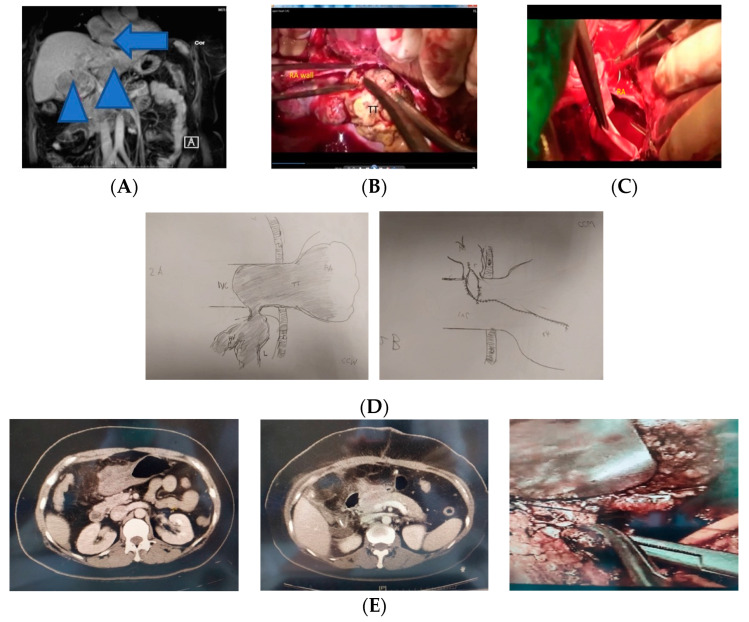
(**A**) Abdominal MRI sagittal view: hepatocellular carcinoma (arrowhead) over the liver with tumor thrombi (arrow) over right atrium (RA). (**B**) After opening the right atrium (RA) wall, the tumor thrombi (TT) spilled out. (**C**) After thrombectomy, the right atrium (RA) lumen was empty. (**D**) The First Operation. (**Left**) Tumor thrombi invading to RA and IVC via HV. (**Right**) Wound closed in continuous suture (RA: right atrium; TT: tumor thrombi; IVC: inferior vena cava; L: liver; HV: hepatic vein; and D: diaphragm). (**E**) Staged Hepatectomy: (**Left**) congested vessels (before first operation). (**Middle**) disappeared congested vessels (before second operation). (**Right**) The liver parenchyma was transected using the Kelly clamp-crush method.

**Figure 2 jcm-11-02140-f002:**
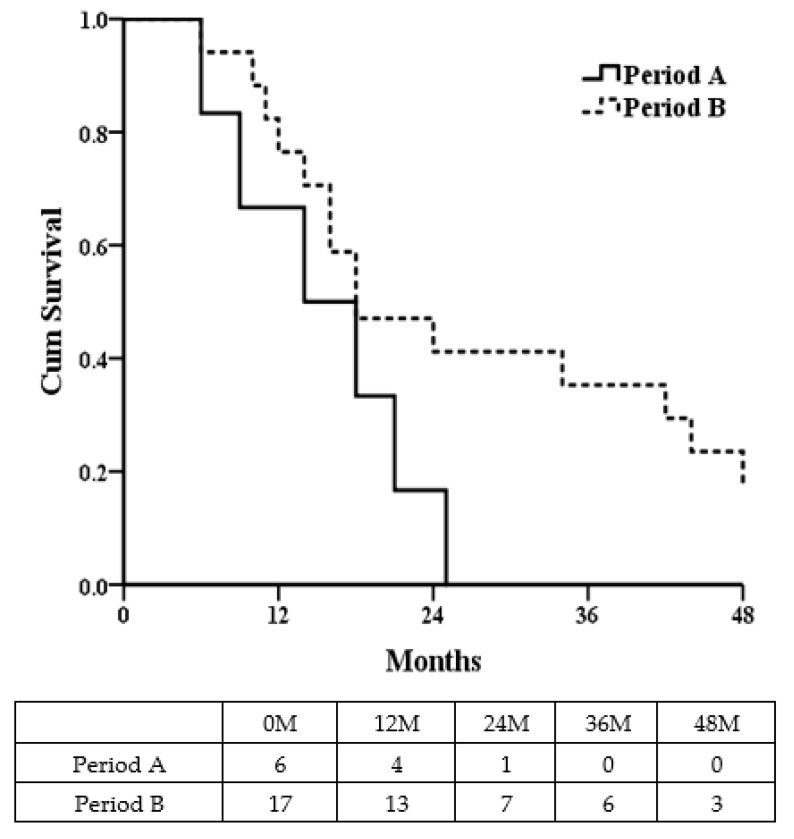
The median overall-survial time (MOST) in different operative periods.

**Figure 3 jcm-11-02140-f003:**
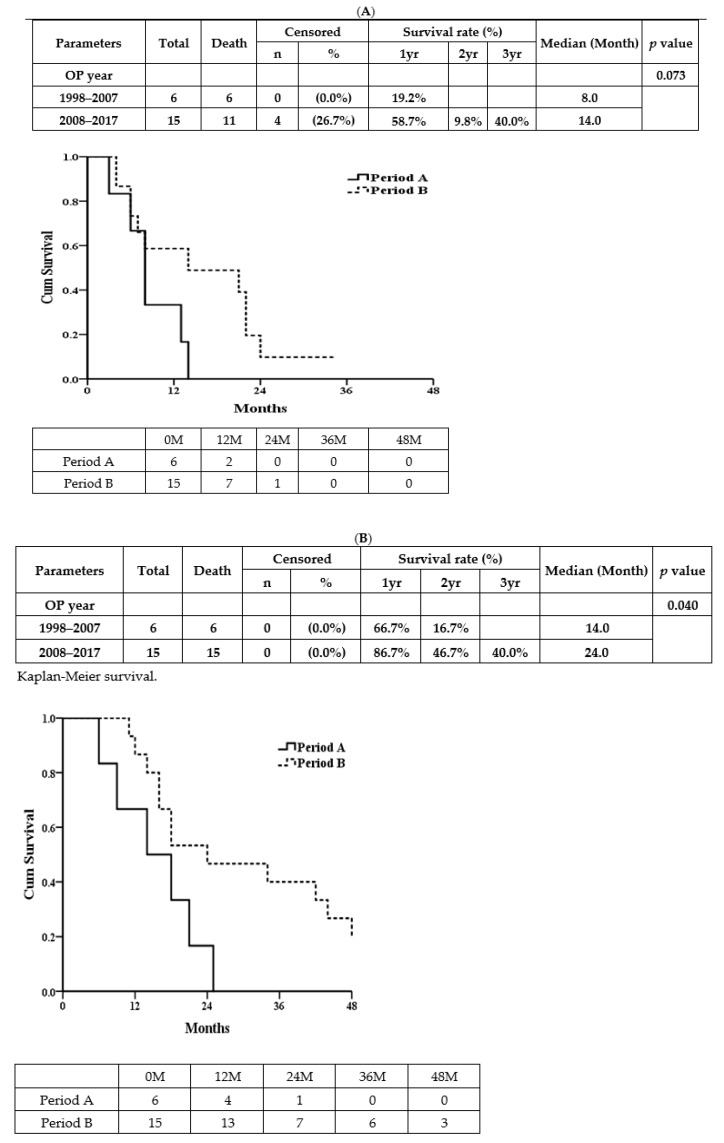
(**A**) The median disease-free survival time (MDFT). (**B**) The median overall-survial time (MOST).

**Table 1 jcm-11-02140-t001:** Clinicopathological characteristics.

	Period A (*n* = 7)	Period B (*n* = 17)	*p* Value
sex (M:F)	6:1	14:3	1.000
age (years)	58 (50–66)	59 (49–65)	0.928
cirrhosis (yes:no)	3:4	10:7	0.659
newly diagnosed: recurrent	5:2	13:4	0.878
hepatitis status B:C:B+C	6:1:0	12:4:1	0.682
serum AFP (ng/mL)	218 (5–11,200)	371 (11–10,411)	0.711
ICGR15 (%)	19.2 (8.0–62.5)	18.6 (7.5–43.4)	0.620
Child-Pugh Grade A:B	6:1	2:15	1.000
main tumor number ≥2	2	5	1.000
tumor size (cm)	6.0(3–9)	5.0 (4–9.5)	0.855
satellite nodule (yes:no)	7:0	17:0	1.000
tumor capsule formation	6	14	1.000
tumor differentiationmoderate:poor	1:6	2:15	1.000

Note: Data are patient number or median (range). B+: seropositive for HBsAg. C+: seropositive for anti-HCV. B+C+: seropositive for both HBsAg and anti-HCV. ICG R15: indocyanine-green 15 min retention rate; ICU: intensive care unit; and AFP: α-fetoprotein.

**Table 2 jcm-11-02140-t002:** Early postoperative results.

	Period A (*n* = 7)	Period B (*n* = 17) *	*p* Value
liver transection time (min)	26.3 (25.0–44.2)	23.8 (11.5–48.2)	0.892
liver transection area (cm^2^)	30.8 (29–47.6)	28.5 (18.0–45.5)	0.286
CPB duration (min)	544.5 (14.5–105)	40.5 (12–102.8)	0.372
operation time (r)	10.3 (9.3–12.3)	9.5 (7.5–10.8) ^#^	0.114
operative blood loss (mL)	6750 (5600–12,800)	1680 (910–8600) ^#^	<0.001 ^#^
blood transfusion (mL)	5500 (2300–11,000)	0 (0–7800) ^#^	<0.001 ^#^
postoperative ICU stay (days)	7 (3–28)	2 (1–12)	0.035
need blood transfusion	7	6	0.015
postoperative hospital stay (days)	26 (22–61)	25 (21–56) ^#^	0.242
Complications	4	4	0.356
SACS	2	2	1.000
bile leakage	1	0	
intraabdominal hematoma	2	2	1.000
arrhythmia	1	1	1.000
Clavien–Dindo grade >3	1	2	1.000

Note: Data are patient number or median (range). ^#^ Data are shown as the sum of the two operations in period 2. SACS: secondary abdominal compartment syndrome. * Two patients did not receive an operation.

## Data Availability

The data generated and analyzed in this study are included in this article.

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
