# Peer review of "Concomitant Hepatectomy and Atrial Thrombectomy under Cardiopulmonary Bypass versus Staged Hepatectomy in the Treatment for Hepatocellular Carcinoma with Large Right Atrial Tumor Thrombi"

_jcm, 2022, doi:10.3390/jcm11082140_

Round 1

Reviewer 1 Report

Chao et al. proposed an interesting paper concerning a rare presentation of HCC, with right atrial tumor thrombosis.

It explains a low number of patients and I am not sure that other teams have got a largely more important experience in this field. That explains probably the bias of this comparison, based on retrospective data, and two large different periods.

Nevertheless, despite this important limit, I just underlined some minor revisions:

  • I am not english good speaker but some mistakes are present and the text must be read cautiously.
  • In summary, the numbers before chapters are unuseful.

Author Response

point 1. It explains a low number of patients and I am not sure that other teams have got a largely more important experience in this field. That explains probably the bias of this comparison, based on retrospective data, and two large different periods.

Response 1: "Please see the table 1."--this is the table i have made. I have had collected the published articles from 1990 to 2020. And one of the columns shows the patient number in each article. We have the most patient numbers so far. But still, as review's comment, we have only 24 patients, not a large number, this is one of our limits.

point 2. I am not english good speaker but some mistakes are present and the text must be read cautiously.

Response 2: Thank you. I will be seeking for extensive English language editing.

point 3. In summary, the numbers before chapters are unuseful.

Response 3: i have removed the numbers before chapters. Thank you for your advice

Reviewer 2 Report

Chao et al. introduced the efficacy and feasibility of their approach for advanced hepatocellular carcinoma (HCC), with tumor thrombi largely extended to right atrium. They recommended staged hepatectomy combined with the first thrombectomy over concomitant hepatectomy and thrombectomy under cardiopulmonary bypass. Their approach is unique, and the paper is generally well written and easy to read. However, as the authors also mentioned, the situation is rare, so that it might be difficult for readers to understand the actual procedure. And, although the authors mentioned the first thrombectomy could reduce the congested retroperitoneal collateral vessels and swollen liver, it should be shown with imaging study before the second surgery of hepatectomy. Anyway, the authors should show the actual cases with informative imaging studies and intraoperative pictures more, to help the readers to understand the actual procedure of this complex situation. Also, the authors should discuss the current approach of advanced chemotherapy including immune-checkpoint inhibitors. It might be adoptable between the first thrombectomy and the second surgery to improve the outcome.

Author Response

point 1. although the authors mentioned the first thrombectomy could reduce the congested retroperitoneal collateral vessels and swollen liver, it should be shown with imaging study before the second surgery of hepatectomy.

Response 1: "Please see the attachment, Figure 1."

point 2. the authors should show the actual cases with informative imaging studies and intraoperative pictures more, to help the readers to understand the actual procedure of this complex situation.

Response 2: "Please see the attachment, Figure 2"

point 3. the authors should discuss the current approach of advanced chemotherapy including immune-checkpoint inhibitors.

Response 3: Hepatic arterial infusion chemotherapy (HAIC) with regimen of Cisplatin, Mitomycin, Fluorouracil and Leucovorin Teva...etc.

Systemic chemotherapy with regimen of Cisplatin, Lipo-Dox and Fluorouracil.

Target therapy, such as Regorafenib, Lenvatinib...etc.

Immunotherapy. since 2017, the U.S. Food and Drug Administration (FDA) approved nivolumab for previous treated HCC with sorafenib. Immunotherapy combination therapy with targeted agent, such as tyrosine kinase inhibitors, radiotherapy, TACE and radiofrequency thermal ablation has shown well anti-cancer efficiency but their objective response rate is still relatively low.

This, serves as an additional therapeutic choice, if sorafenib is ineffective.
